# Unveiling Species Diversity Within Early-Diverging Fungi from China XI: Eight New Species of *Cunninghamella* (*Mucoromycota*)

**DOI:** 10.3390/microorganisms13112508

**Published:** 2025-10-31

**Authors:** Yang Jiang, Heng Zhao, Xin-Yu Ji, Zi-Ying Ding, Wen-Xiu Liu, Fei Li, Shi Wang, Xiao-Yong Liu, Zhe Meng

**Affiliations:** 1College of Life Sciences, Shandong Normal University, Changqinghu Campus, Jinan 250358, China; jiangyang202309@126.com (Y.J.);; 2CAS Key Laboratory of Forest Ecology and Silviculture, Institute of Applied Ecology, Chinese Academy of Sciences, Shenyang 110016, China; 3Institute of Microbiology, Chinese Academy of Sciences, Beijing 100101, China

**Keywords:** *Cunninghamella*, basal fungi, fungal diversity, taxonomy, molecular phylogeny

## Abstract

The genus *Cunninghamella* is widely distributed, primarily saprotrophic, occasionally endophytic and phytopathogenic. Analysis based on the internal transcribed spacer (ITS), the large subunit (LSU) of ribosomal DNA, and translation elongation factor 1 alpha (*TEF1α*), along with morphological comparisons, resulted in a discovery of eight new species. Molecular phylogenetic analyses placed each of these new species within well-supported clades. *Cunninghamella crassior* sp. nov., with short and thick spines, and *C. fusca* sp. nov. with brown sporangiola, are sister clades to each other. *C. diffundens* sp. nov., containing dispersed granules in sporangiola, is closely related to *C. irregularis* Zhao. *C. tuberculata* sp. nov., producing sporangiola with nodule-like protrusions. *C. fulvicolor* sp. nov., sister to *C. irregularis*, forms yellowish-brown pigmented colonies. *C. guttulata* sp. nov., with teardrop-shaped sporangiola, and *C. inaequalis* sp. nov., with uneven sporangiola, are both closely related to *C. regularis* Zhao. *C. monosporangiola* sp. nov., characterized by only one sporangiolum on some vesicles, is sister to *C. verrucosa* Zhao. This study represents the eleventh installment in a series investigating early-diverging fungal diversity in China and expands the number of accepted species in *Cunninghamella* to 39.

## 1. Introduction

*Cunninghamella* Matruchot belongs to *Cunninghamellaceae*, *Mucorales*, *Mucoromycetes*, *Mucoromycota* (https://indexfungorum.org/, accessed on 21 April 2025). The genus exhibits a significant metabolic capacity, characterized by the biosynthesis of a diverse spectrum of compounds including fatty acids, terpenes, sugars, and nickel-iron carriers [1]. Members are distributed worldwide, with a rich record in Australia, Brazil, Cameroon, Canada, China, Czech Republic, India, Latvia, Nigeria, and Thailand (GBIF Backbone Taxonomy. https://doi.org/10.15468/39omei, accessed via https://www.gbif.org/species/2551965, accessed on 17 April 2025) (https://www.gbif.org/, accessed on April 2025) [2,3,4,5,6,7,8,9,10,11,12,13,14].

As the type genus of the family *Cunninghamellaceae*, it was established by Matruchot in 1903 [15,16]. Nearly a century later, Zheng and Chen [2] conducted a monographic study based on morphology and molecular systematics. Since then, many new species of *Cunninghamella* have been introduced based on phylogenetic data and morphological characteristics [2,3,4,5,17,18]. Fungi of the genus *Cunninghamella* exhibit well-developed, broad hyphae that are generally aseptate but may occasionally form septa [6,16,19]. Their colonies are cottony or fluffy, appearing white initially before maturing to gray or dark gray [20]. The sporangiophores end in vesicles that bear single-spored, spinulose (short-spined) sporangiola [7,8,9,10,21].

The genus *Cunninghamella*, within the order *Mucorales*, is primarily differentiated by a combination of morphological characteristics of its asexual stage and molecular markers. Morphologically, key diagnostic features include the development of erect sporangiophores that typically exhibit verticillate branching. These branches terminate in swollen, spherical to pyriform vesicles, which are not true sporangia as they bear denticles (small projections) on their surface. The asexual spores, known as globose to subglobose sporangiospores, are produced directly on these denticles. Species identification relies on specific traits such as the branching pattern of the sporangiophores (e.g., presence of apical vesicles, number and arrangement of branches), vesicle shape (e.g., spherical, polyhedral, or irregular) and size, and spore morphology.

The classification of the genus *Cunninghamella* has long been challenged by the limitations of morphology-based approaches. Although the integration of morphological observations with molecular phylogenies (e.g., using ITS rDNA and TEF1α sequences) by Liu and Zhao [17,22] established a robust consensus framework, the true diversity within this genus, particularly in under-explored regions like southern China, remains poorly understood. It is plausible that several undiscovered species, potentially forming species complexes, constitute a hidden diversity awaiting revelation under this modern framework. To address this gap, we applied this integrated approach to the investigation of soil fungi from southern China. This study, which is the eleventh installment in our series on early-diverging fungi in China [23,24,25,26,27,28,29,30,31,32], aims to delineate and formally describe the novel *Cunninghamella* species discovered, thereby enriching our comprehension of the genus’s diversity and phylogeny.

## 2. Materials and Methods

### 2.1. Isolation and Morphology

Soil samples were obtained from Hainan Province in April and October 2023, and from Yunnan Province in June 2023. Fungal strains were subsequently isolated from these samples using a combination of the soil dilution and single spore methods. A soil suspension with a concentration of 10^−1^ was prepared by shaking approximately 1 g of soil sample with 10 mL of sterile water. Subsequently, 1 mL of this diluted suspension was introduced into 9 mL of fresh sterile water to generate a 10^−2^ concentration suspension. Soil suspensions with 10^−3^ and 10^−4^ concentrations were prepared using the same serial dilution method as above [23,24,25,26,27,28,29,30,33]. The soil suspension (200 μL) was evenly spread on Rose Bengal Chloramphenicol (RBC: peptone 5.00 g/L, glucose 10.00 g/L, KH_2_PO_4_ 1.00 g/L, MgSO_4_·7H_2_O 0.50 g/L, rose red 0.05 g/L, chloramphenicol 0.10 g/L, agar 15.00 g/L) plates using a sterile L-shaped spreader, and incubated in darkness at 25 °C for 2–5 d. The mycelium was transferred from the colonial margin to a fresh Potato Dextrose Agar (PDA: glucose 20 g/L, potato 200 g/L, agar 20 g/L) medium. Photographs of both the obverse and reverse of the colony were captured using a digital camera (Canon PowerShot G7X, Canon, Tokyo, Japan). All strains were stored with 10% sterilized glycerol at 4 °C. Microscopic morphology was observed using an optical microscope (Olympus BX53 Manufacturer: Olympus Corporation Tokyo, Japan) and captured with a high-definition digital camera (Olympus DP80 Manufacturer: Olympus Corporation Tokyo, Japan) [34,35,36,37]. Morphological traits were measured using Digimizer software (v5.6.0), each with a minimum of 30 individuals. The living cultures were preserved at the China General Microbiological Culture Collection Center, Beijing, China (CGMCC). Equivalent strains were stored at Shandong Normal University (XG). Dry cultures of types were kept at the Herbarium Mycologicum Academiae Sinicae, Beijing, China (Fungarium; HMAS). The taxonomic information was deposited at the Fungal Names repository (https://nmdc.cn/fungalnames/ (accessed on 24 October 2025)).

### 2.2. DNA Extraction and Amplification

DNA was extracted using an extraction kit (Changchun GeneOn BioTech Co., Ltd., Changchun, China). The ITS, LSU, and *TEF1α* regions were amplified using the primers and programs listed in Table 1. The PCR mixture (25 μL final volume) was prepared using 12.5 μL of 2 × Hieff Canace Plus PCR Master Mix with dye (Yeasen Biotechnology, Cat No. 10154ES03. Shanghai, China), to which 9.5 μL of ddH_2_O, 1 μL of each primer (10 μM), and 1 μL of genomic DNA template (1 ng/μL) were added. PCR products were subjected to 1% agarose gel electrophoresis and observed under UV light (Shenhua Science Technology Co., Ltd., Hangzhou, China) at 254 nm [38,39].

### 2.3. Phylogenetic Analyses

Sequences were obtained from a nucleotide database from NCBI (see Table 2) [4]. *Mucor janssenii* Lendn was chosen as the outgroup for its close phylogenetic affinity to *Cunninghamella*. This single, closely-related outgroup served as an unambiguous root for the ingroup analysis. Newly acquired sequences were processed with MEGA v.7.0 to ensure alignment consistency [44]. Phylogenetic analyses on single markers and the ITS-LSU-*TEF1α* combined matrix were conducted to reconstruct the phylogeny, employing both ML and BI approaches integrated into the CIPRES Science Gateway V.3.3 (https://www.phylo.org/, accessed on 15 February 2025). Maximum Likelihood (ML) analyses were carried out with 1000 bootstrap replicates using RAxML 8.2.4 (https://www.phylo.org/, accessed on 15 February 2025), and Bayesian Inference (BI) was performed under the GTR + I + G model with a sampling frequency of 1000 generations. The substitute model is calculated using PAPU software (version 4.0b10), and the relevant parameters are set to DSet distance = JC objective = ME base = equal rates = equal pinv = 0subst = all negbrlen = setzero; NJ showtree = no breakties = random. After running MCMC analyses on concatenated genes for 835,000 generations, we computed posterior probabilities from the sampled trees to construct majority-rule consensus trees. Eight cold Markov chains were executed simultaneously across two million generations, ensuring MCMC convergence criteria [45,46]. The resulting phylogenetic trees underwent iTOL based optimization [47], followed by visual refinement in Adobe Illustrator CC 2019.

## 3. Result

### 3.1. Molecular Phylogeny

We performed phylogenetic analyses of *Cunninghamella* using a dataset composing 73 strains of 40 species, with *Mucor janssenii* (CBS 205.68) designated as outgroup. The concatenated sequence matrix contains 3157 aligned characters, partitioned as 1–1514 positions (ITS), 1515–2591 (LSU), and 2592–3157 (*TEF1α*). Three categories of sites were identified: 1152 parsimony-informative, 551 parsimony-uninformative, and 1454 constants. Bayesian analyses employed with the GTR+G model for ITS and the GTR+I+G model for LSU and *TEF1α* loci. The Bayesian phylogeny exhibits topological congruence with the ML tree, supported by concordant nodal bootstrap as well as BI posterior probability values. The phylogenetic analysis showed eight monographic clades (Figure 1), and eight novel species were consequently identified within *Cunninghamella*. They are *C. diffundens* sp. nov., *C. crassior* sp. nov., *C. fulvicolor* sp. nov., *C. fusca* sp. nov., *C. guttulata* sp. nov., *C. inaequalis* sp. nov., *C. monosporangiola* sp. nov., and *C. tuberculata* sp. nov.

### 3.2. Taxonomy

#### 3.2.1. *Cunninghamella Crassior* Y. Jiang, H. Zhao, Z. Meng & X.Y. Liu. sp. nov. Figure 2

Fungal Name Number: FN 571687

Etymology—The epithet “*crassior*” refers to thicker spines.

Type—China, Yunnan Province (27.4375858° N, 99.809103° E), soil sample, collected by H. Zhao, holotype HMAS 353832, ex-holotype CGMCC3.28882

Description—Colonies on PDA, initially white becoming Drab Gray, attained dimensions of 70 mm in diameter and 10 mm in height when cultured at 27 °C for 6 days, exhibiting a floccose texture and a regular reverse margin. The hyphae were branched, mostly aseptate when young (4.5–20.5 μm in diam.), with rhizoids abundant and stolons present. Sporangiophores developing from these were erect, straight or bent, and unbranched or verticillate, broadening upwards and lacking septa. Their vesicles were subglobose to clavate, hyaline, and rough, measuring 7.0–36.5 μm long by 5.0–29.5 μm wide, and carried spiny, globose and brownish sporangiola (7.5–13.0 μm) on pedicels of 3.0–4.5 μm. No chlamydospores or zygospores were detected.

**Figure 2 microorganisms-13-02508-f002:**
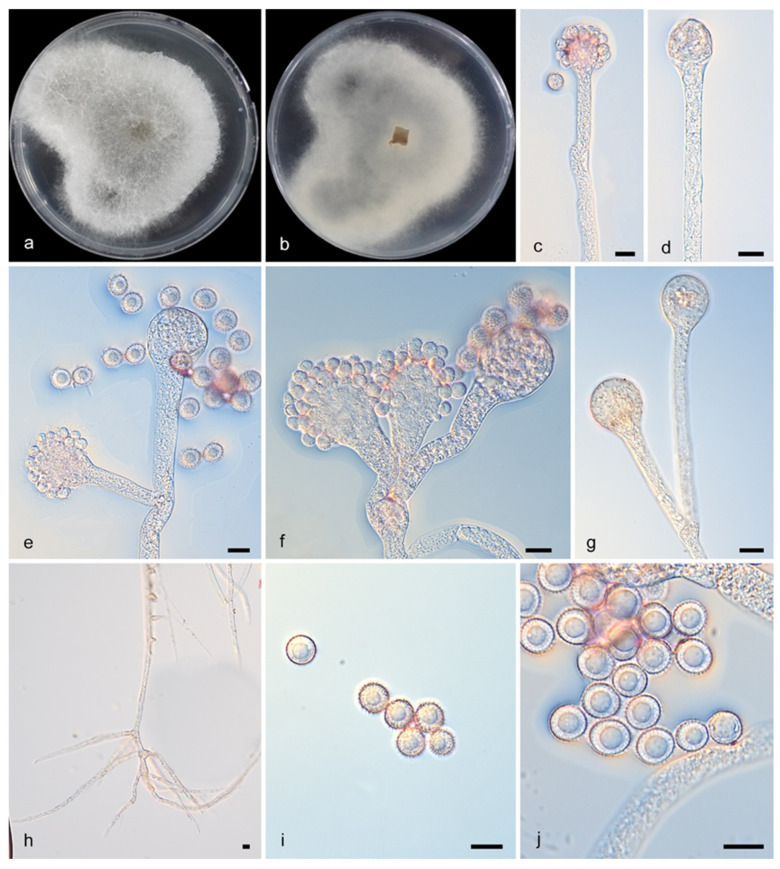
*Cunninghamella crassior* (CGMCC3.28882). (**a**,**b**): Colonies on PDA after 7 days of cultivation ((**a**), obverse, (**b**), reverse); (**c**–**g**): Sporangiophores showing branching patterns; (**h**): Rhizoids; (**i**,**j**): Sporangiola. Scale bars: 10 µm (**c**–**j**).

Notes—*Cunninghamella crassior* (CGMCC3.28882) was isolated from a soil sample in Yunnan Province, China. In terms of phylogeny, *C. crassior* (CGMCC3.28882) clusters alone on a separate branch. Compared with the ITS sequence of *C. clavata* Zheng (CBS 362.95), the similarity is 96.86% (693/728 identical, 45/591 gaps). The similarity of the LSU sequence is 100% (776/777 identical, 0/777 gaps), and the similarity of the TEF1α sequence is 99.8% (560/567 identical, 0/567 gaps). Morphologically, *C. crassior* (CGMCC3.28882) releases sporangiospores with short thick spines, and no similar morphology has been found.

#### 3.2.2. *Cunninghamella diffundens* Y. Jiang, H. Zhao, Z. Meng & X.Y. Liu. sp. nov. Figure 3

Fungal Name: FN573054

Etymology—The epithet “*diffundens*” refers to dispersed granules in sporangiola.

Type—China, Yunnan Province (26.5642168° N, 101.268054° E), soil sample, collected by H. Zhao, holotype HMAS 353829, ex-holotype CGMCC 3.28881.

**Figure 3 microorganisms-13-02508-f003:**
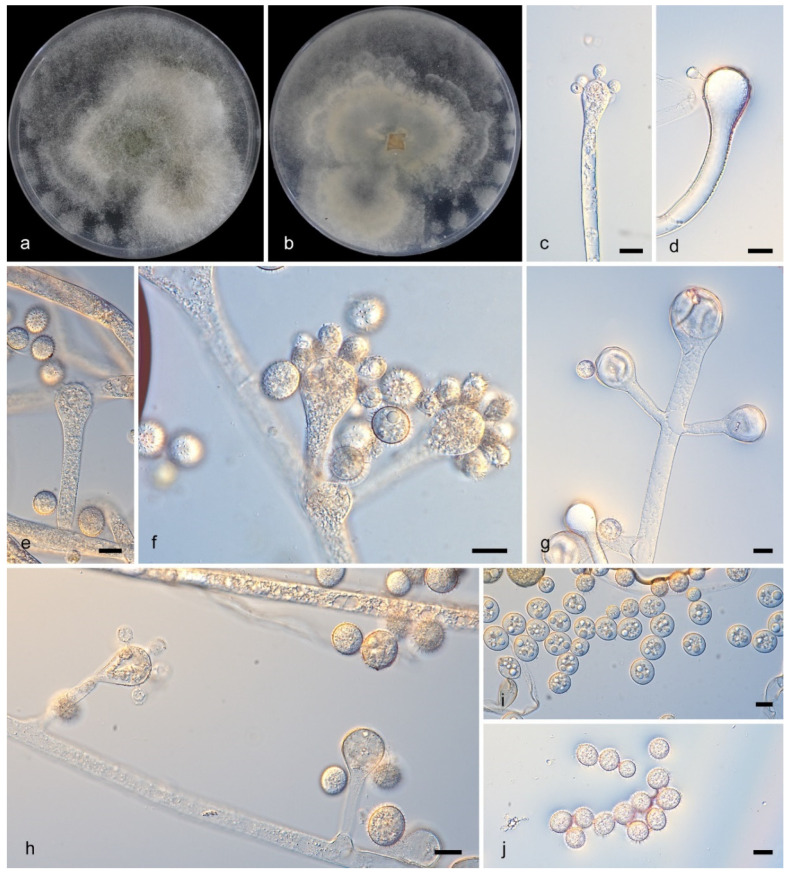
*Cunninghamella diffundens* (ex-holotype CGMCC 3.28881).(**a**,**b**): Colonies on PDA after 7 days of cultivation ((**a**), obverse; (**b**), reverse); (**c**–**h**): Sporangiophores showing branching patterns; (**i**–**j**): Sporangiola, showing spines and dispersed granules. Scale bars: 10 µm (**c**–**j**).

Description—Colonies on PDA at 27 °C for 7 days, reaching 90 mm in diameter, 10–15 μm high, initially white, soon becoming Drab Gray, floccose, in reverse irregular at margin. Hyphae branched, aseptate when young, septate when old, 5.4–14.5 μm in diameter. Rhizoids present. Stolons present. Sporangiophores arising from stolons or aerial hyphae, erect, straight or recumbent, sometimes slightly broadening upwards in main axes, solitary or branched in pairs, sympodially or 2–5 verticillate, 21.5–171.5 μm long. Sporangiola containing dispersed granules, borne on pedicels on vesicles, spiny, mostly ovoid, 8.8–15.1 μm in length and 8.0–12.8 μm in width. Chlamydospores absent. Zygospores unknown.

Notes—The holotype strain *Cunninghamella diffundens* (CGMCC3.28881), collected from a soil sample in Yunnan Province, China, forms a distinct monophyletic clade in phylogenetic analyses. It shows close genetic affinity to *C. irregularis* (CGMCC 3.16113), with sequence similarities of 95.8% in the ITS region (765/790 identical sites, 8 gaps) and 99.8% in the LSU region (940/942 identical sites, 1 gap). Notably, that no *TEFα* sequence of *C. irregularis* (CGMCC 3.16113) was available for comparison. Morphologically, *C. diffundens* (CGMCC3.28881) produces two distinct types of sporangiola, a feature unobserved in related species. *C. diffundens* (CGMCC3.28881) was described as a new species.

#### 3.2.3. *Cunninghamella fulvicolor* Y. Jiang, H. Zhao, Z. Meng & X.Y. Liu. sp. nov. Figure 4

Fungal name: FN573067

Etymology—The epithet “*fulvicolor*” refers to the yellowish-brown pigmentation of colonies.

Type—China, Yunnan Province (26.9523432° N, 99.9502813° E), soil sample, collected by H. Zhao. Holotype HMAS 353825, ex-holotype CGMCC3.28884.

Description—Colonies on PDA at 25 °C for 5 days, reaching 90 mm in diameter, more than 20 mm high, initially white, gradually becoming light gray, floccose, produce dark or brown precipitate, irregular at margin. Hyphae flourishing, branched, aseptate when young, septate when old, 5.7–9.5 μm in diameter. Rhizoids infrequent, root-like, always branched. Stolons present. Sporangiophores developed erectly and straight from aerial hyphae, consistently broadening upwards; they were monopodial, appearing either unbranched or simply branched in pairs but never verticillate; septa were frequently present in the upper parts. The vesicles were subglobose to globose, generally hyaline or subhyaline (10.0–19.2 μm long, 6.9–17.4 μm wide), though occasionally brownish. Attached via pedicels, the sporangiola were ovoid, hyaline, and measured 9.0–12.5 μm by 4.7–7.4 μm, often exhibiting droplets in youth. Chlamydospores absent. Zygospores unknown.

Notes—The holotype strain of *Cunninghamella fulvicolor* was isolated from soil in Yunnan Province, China. Phylogenetic analyses revealed that *C. fulvicolor* (CGMCC3.28884) and *C. fulvicolor* (HZ012-2) represent the same strain, with maximum likelihood (ML) and Bayesian inference (BI) support values of 100/1. Both *C. fulvicolor* (CGMCC3.28884) and *C. fulvicolor* (HZ012-2) clustered closely with *C. irregularis* (CGMCC 3.16113) on the same branch, indicating a close genetic relationship. However, morphological differences were observed: *C. irregularis* (CGMCC 3.16113) produces irregular spores: (with thick spines, ovoid, 10.0–12.5 μm long and 8.0–10.5 μm wide, and globose, 9.0–16.5 μm in diameter.) *C. fulvicolor* produces only one morphotype of sporangiola (length: 9.0–12.5 μm, width: 4.7–7.4 μm). Conclusion: based on morphological distinctiveness and molecular phylogenetic divergence, *C. fulvicolor* and *C. irregularis* (CGMCC 3.16113) are recognized as two distinct species.

**Figure 4 microorganisms-13-02508-f004:**
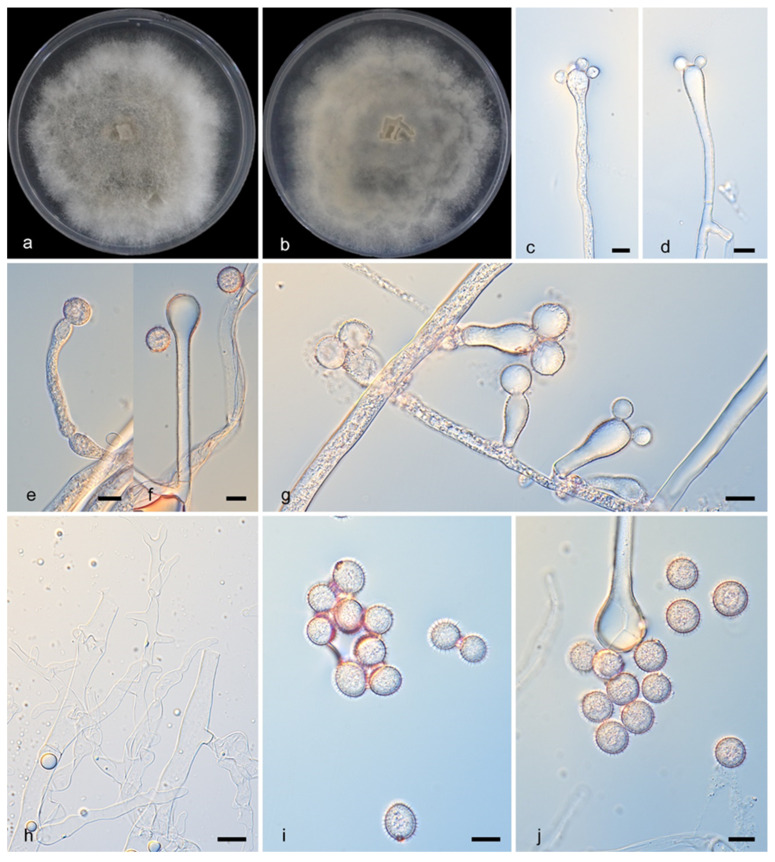
*Cunninghamella fulvicolor* (CGMCC3.28884). (**a**,**b**): Colonies on PDA after 5 days of cultivation ((**a**) obverse, (**b**) reverse); (**c**–**g**): Sporangiophores showing branching patterns; (**h**): Rhizoids; (**i**,**j**): Sporangiola. Scale bars: 10 µm (**c**–**j**).

#### 3.2.4. *Cunninghamella fusca* Y. Jiang, H. Zhao, Z. Meng & X.Y. Liu. sp. nov. Figure 5

Fungal name: FN573068

Etymology—The epithet “*fusca*” describes the brown pigmentation of sporangiola.

Type—China, Yunnan Province (26.9523432° N, 99.9502813° E), soil sample, collected by H. Zhao. Holotype HMAS 353826, ex-holotype CGMCC3.28885.

Description—Colonies on PDA incubated at 27 °C for 7 days attained a diameter of 80 mm and a height of 10 mm; they were floccose, light grayish-white in color, and exhibited an irregular margin. The mycelial system consisted of flourishing, branched hyphae (2.0–9.6 μm in diam.) that developed septa with age, and produced abundant, often swollen, root-like rhizoids along with stolons. The reproductive structures featured sympodially branched sporangiophores (37.0–107.5 μm long) that arose aerially, were typically straight, and always broadened upwards; these bore vesicles supporting large, globose, and thick-spined sporangiola (14.1–35.5 μm in diam.). Chlamydospores absent. Zygospores unknown.

**Figure 5 microorganisms-13-02508-f005:**
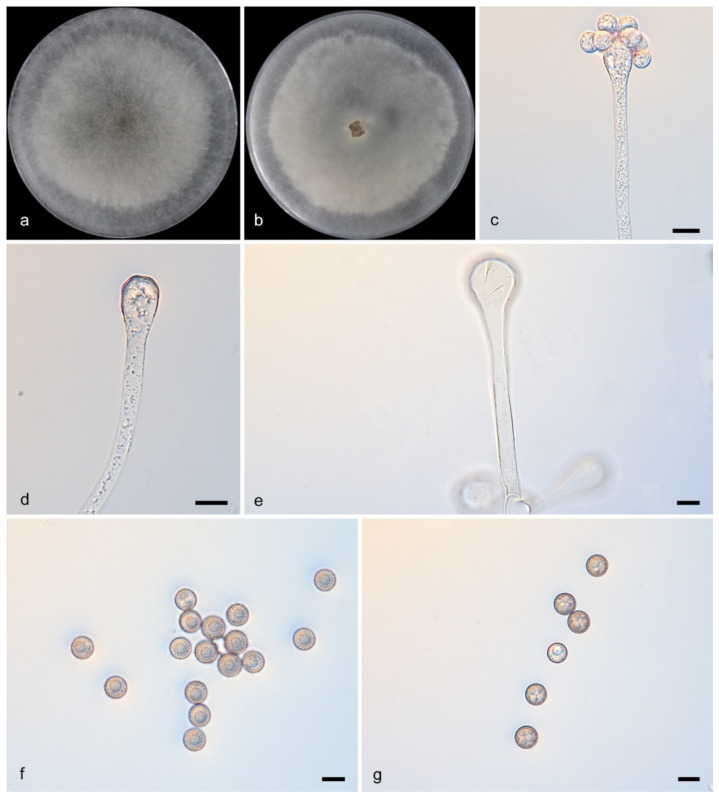
*Cunninghamella fusca* (CGMCC3.28885). (**a**,**b**): Colonies on PDA after 7 days of cultivation ((**a**), obverse; (**b**), reverse); (**c**–**e**): Sporangiophores showing branching patterns; (**f**,**g**): Sporangiola. Scale bars: 10 µm (**c**–**g**).

Notes—Strains CGMCC 3.28885 and HZ108-2 form a robust clade (MLBS/BIPP = 100/1), sister to *Cunninghamella crassior* (Figure 1). Morphologically, these two strains produce brown sporangiola, while *C. crassior* produce brownish ones.

#### 3.2.5. *Cunninghamella guttulata* Y. Jiang, H. Zhao, Z. Meng & X.Y. Liu. sp. nov. Figure 6

Fungal name: FN573069

Etymology—The epithet “*guttulata*” refers to teardrop-shaped sporangiola.

Type—China, Hainan Province, Lingshui Li Autonomous County (18.668825° N, 109.919678° E), soil sample, collected by Y.X. Wang. Holotype HMAS 353833, ex-holotype CGMCC3.28886.

**Figure 6 microorganisms-13-02508-f006:**
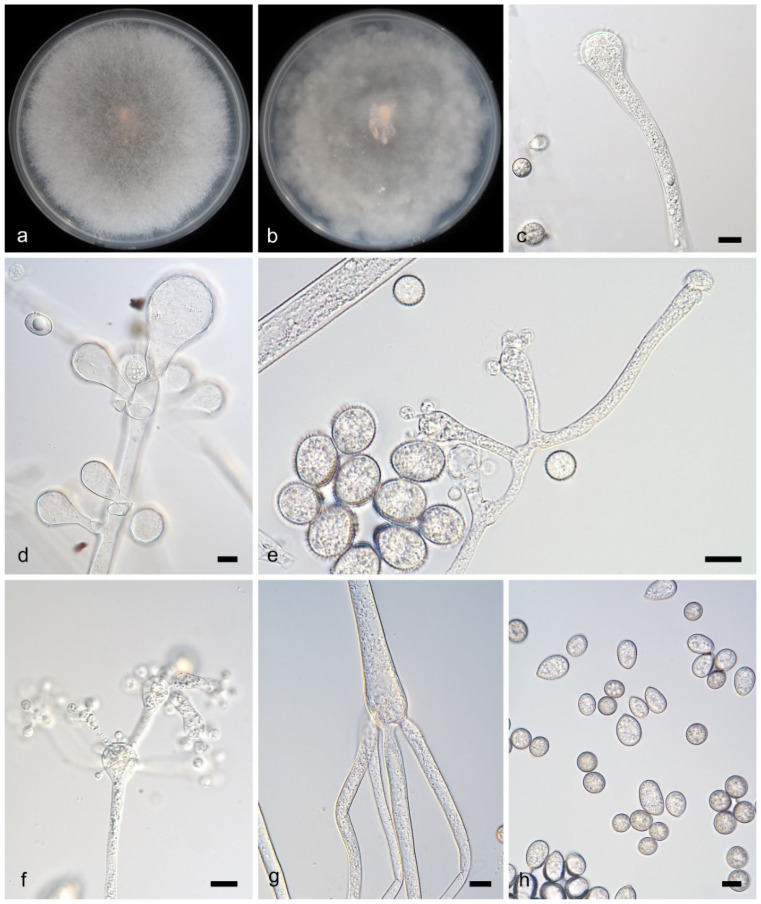
*Cunninghamella guttulata* (CGMCC3.28886). (**a**,**b**): Colonies on PDA after 5 days of cultivation ((**a**), obverse; (**b**), reverse); (**c**–**f**): Sporangiophores showing branching patterns; (**g**): Rhizoids; (**h**): sporangiola. Scale bars: 10 µm (**c**–**h**).

Description—Colonies on PDA at 27 °C for 5 days, reaching 90 mm in diameter, 15 mm high, initially white, becoming Drab Gray with age, floccose, in reverse regular at margin. Hyphae branching, aseptate when young, rarely septate with age, 4.5–20.5 μm in diameter. Rhizoids abundant, root-like, branched. Stolons present. Sporangiophores developed from stolons or aerial hyphae. They were erect, either straight or bent, and exhibited an unbranched or verticillate (2–6 branches) architecture, consistently broadening upwards with frequent septation in the upper regions. The terminal vesicles were subglobose to globose, generally hyaline or subhyaline, and measured 13.5–40.5 μm in length and 12.5–40.5 μm in width. Attached via pedicels, the sporangiola were ovoid and eardrop-shaped (9.0–12.5 × 7.5–9.0 μm), hyaline to subhyaline, often bearing droplets in their early stages, and characterized by a distinctive ornamentation of ovate or drop-shaped spines. Chlamydospores absent. Zygospores unknown.

Notes—The holotype strain of *Cunninghamella guttulata* (CGMCC3.28886) was isolated from a soil sample collected in Lingshui Li Autonomous County, Hainan Province, China. Phylogenetically, *C. guttulata* (CGMCC3.28886) and *C. guttulata* (XG04011-1-2) form a distinct clade with strong statistical support (BI/ML = 94/1). Comparative sequence analysis with *C. regularis* (CGMCC 3.16114) revealed: ITS similarity: 97.4% (607/618 identical sites, 18/618 gaps); LSU similarity: 99.7% (925/925 identical sites, 0/925 gaps); *TEF1α* sequence: Not available. Morphologically, *C. guttulata* (CGMCC3.28886) produces teardrop-shaped sporangiola (length: 9.0–12.5 μm, width: 7.5–9.0 μm), a feature not observed in *C. regularis* (CGMCC 3.16114), which produces spherical sporangiola (diameter: 6.5–10.0 μm).

#### 3.2.6. *Cunninghamella inaequalis* Y. Jiang, H. Zhao, Z. Meng & X.Y. Liu. sp. nov. Figure 7

Fungal name: FN573070

Etymology—The epithet “inaequalis” refers to the size variation of sporangiola.

Type—China, Yunnan Province (26.5642168° N, 101.268054° E), soil sample, collected by H. Zhao. Holotype HMAS 353831, ex-holotype CGMCC3.28888.

**Figure 7 microorganisms-13-02508-f007:**
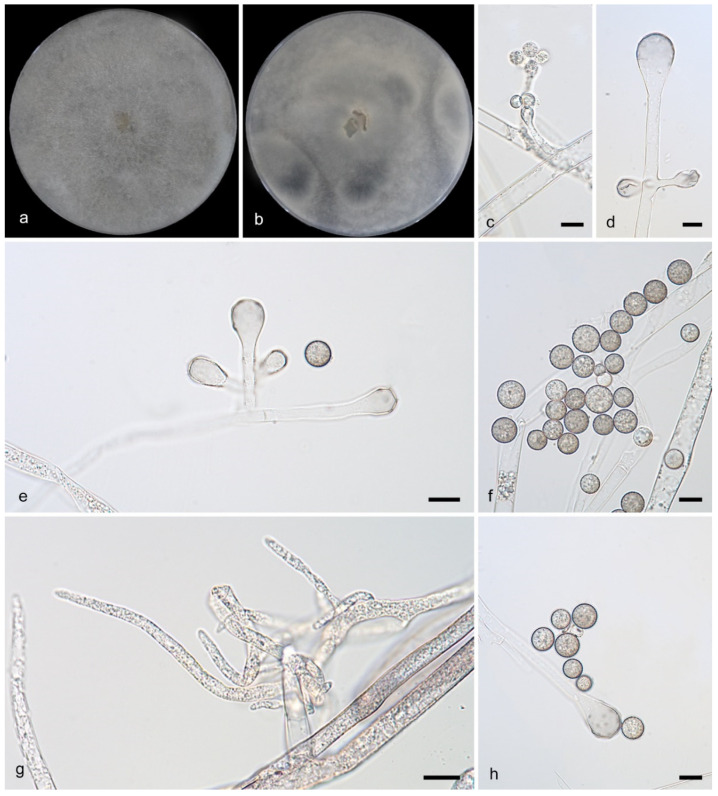
*Cunninghamella inaequalis* (CGMCC3.28887). (**a**,**b**): Colonies on PDA after 7 days of cultivation ((**a**), obverse; (**b**), reverse); (**c**–**e**): Sporangiophores showing branching patterns; (**g**): Rhizoids; (**f**,**h**): Sporangiola. Scale bars: 10 µm (**c**–**h**).

Description—Colonies on PDA at 27 °C for 7 days, fast growing, reaching 90 mm in diameter, 20 mm high, at first white, finally Light Gray. Hyphae flourishing, branched, aseptate when young, septate with age, 4.5–17.0 μm in diameter. Stolons present. Sporangiophores were aseptate and developed erectly and straight from aerial hyphae, consistently broadening upwards; they exhibited an unbranched or simply paired, sympodial/dichotomous branching habit, with ovoid or globose vesicles present on the main stalks. Pedicels 1.5–4.0 μm long. Sporangiola borne on pedicels on vesicles, globose, 10.0–17.0 μm in diameter, hyaline when young, Light Brown with age, with many thin spines on the surface. Chlamydospores absent. Zygospores unknown.

Notes—The holotype strain *Cunninghamella inaequalis* (CGMCC3.28887) was isolated from a soil sample in Yunnan Province, China. Phylogenetically, *C. inaequalis* (CGMCC3.28887) forms a distinct clade. Comparative analysis with *C. regularis* (CGMCC 3.16114) revealed 97.5% ITS sequence similarity (578/591 identical, 14/591 gaps) and 100% LSU sequence similarity (1439/1439 identical, 0/1439 gaps); the *TEF1α* sequence of *C. regularis* was not available. Morphologically, *C. inaequalis* produces smaller sporangiola (4.4–12.6 μm in diameter) compared to *C. regularis* (6.5–10.0 μm in diameter). Additionally, *C. regularis* sporangiola exhibit slight spines, whereas *C. inaequalis* sporangiola are smooth with no spines observed.

#### 3.2.7. *Cunninghamella monosporangiola* Y. Jiang, H. Zhao, Z. Meng & X.Y. Liu. sp. nov. Figure 8

Fungal name: FN573071

Etymology—The epithet “*monosporangiola*” refers to single sporangiola.

Type—China, Yunnan Province (26.5642168° N, 101.268054° E), soil sample, collected by H. Zhao, Holotype HMAS 353830, ex-holotype CGMCC 3.28888.

Description—Colonies on PDA at 27 °C for 5 days reached 90 mm in diameter and 10–15 mm in height, were floccose, transitioned from white to Drab Gray, and had an irregular reverse margin. Hyphae were branched and 2.5–15.0 μm wide, aseptate when young but septate when mature. Sporangiophores arose from stolons or aerial hyphae, were erect, straight or recumbent, 1.5–251.5 μm long, sometimes broadening upwards, and branched singly, in pairs, sympodially, or verticillately (2–5), often septate below vesicles. Vesicles were ovoid/globose, mostly hyaline (6.0–44.0 × 5.0–40.0 μm). Sporangiola on pedicels (1.0–1.5 μm) were ovoid/globose (7.0–14.5 × 6.5–12.5 μm), thick-spined and produce single sporangiola (Figure 8e,h). Chlamydospores absent. Zygospores unknown.

Notes—The type strain *Cunninghamella monosporangiola* (CGMCC3.28888) was isolated from a soil specimen collected in Yunnan Province, China. Phylogenetically, *C. monosporangiola* (CGMCC3.28888) formed a distinct clade in the phylogenetic tree and exhibited 54 nucleotide differences compared to *C. verrucosa* (CGMCC 3.16260) (ITS: 49/637, LSU: 4/1442, *TEF1α*: 1/567). Morphologically, *C. monosporangiola* (CGMCC3.28888) was observed to release a single sporangiospore from its vesicles, a feature not documented in other strains.

**Figure 8 microorganisms-13-02508-f008:**
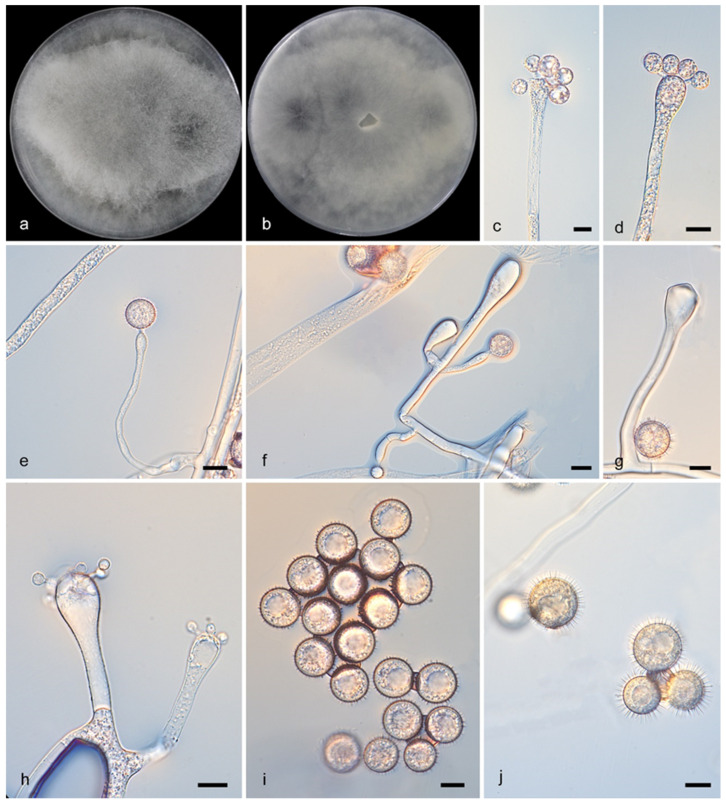
*Cunninghamella monosporangiola* (CGMCC3.28888). (**a**,**b**): Colonies on PDA after 5 days of cultivation ((**a**), obverse; (**b**), reverse); (**c**–**h**): Sporangiophores showing characteristic branching patterns; (**i**,**j**): Sporangiola. Scale bars: 10 µm (**c**–**j**).

#### 3.2.8. *Cunninghamella tuberculata* Y. Jiang, H. Zhao, Z. Meng & X.Y. Liu. sp. nov. Figure 9

Fungal name: FN573072

Etymology—The epithet “*tuberculata*” describes the tuberculate, nodular, wart-like ort knotted protrusions on sporangiola.

Type—China, Yunnan Province (26.5642168° N, 101.268054° E), soil sample, collected by Zhao Heng, Holotype HMAS 353828, ex-holotype CGMCC3.28889.

Description—Colonies on PDA floccose, reaching 9 cm at 25 °C for 5 days, initially white, soon gray, finally gray-brownish, in reverse gray. Hyphae branched, 3.5–19.5 μm diameter, septate when old. Rhizoids and stolons present. Sporangiophores erect or recumbent, arising from stolons or aerial hyphae, main axes usually equivalent and 3.5–20.4 μm in diameter, rarely gradually thickening upwards, primary branches 61.5–287.7 μm long and 4.6–13.4 μm wide, rebranching or third-branching into branchlets. Sporangiola on pedicels on vesicles, spherical, spiny, covered with tuberculate, nodular, wart-like ort knotted protrusions, 4.1–10.7 μm in diameter. Chlamydospores absent. Zygospores unknown.

Notes—The type strain *Cunninghamella tuberculata* (CGMCC3.28889) was isolated from a soil specimen collected in Yunnan Province, China (N: 26.5642168°, E: 101.268054°). Phylogenetically, *C. tuberculata* (CGMCC3.28889) formed a distinct clade in the phylogenetic tree and showed close affinity to *C. guttata* (CGMCC 3.16112), which also occupied an independent clade. Morphologically, *C. tuberculata* produces smooth-walled sporangiola with tuberculate (nodule-like) protrusions, a trait not observed in any known related species.

**Figure 9 microorganisms-13-02508-f009:**
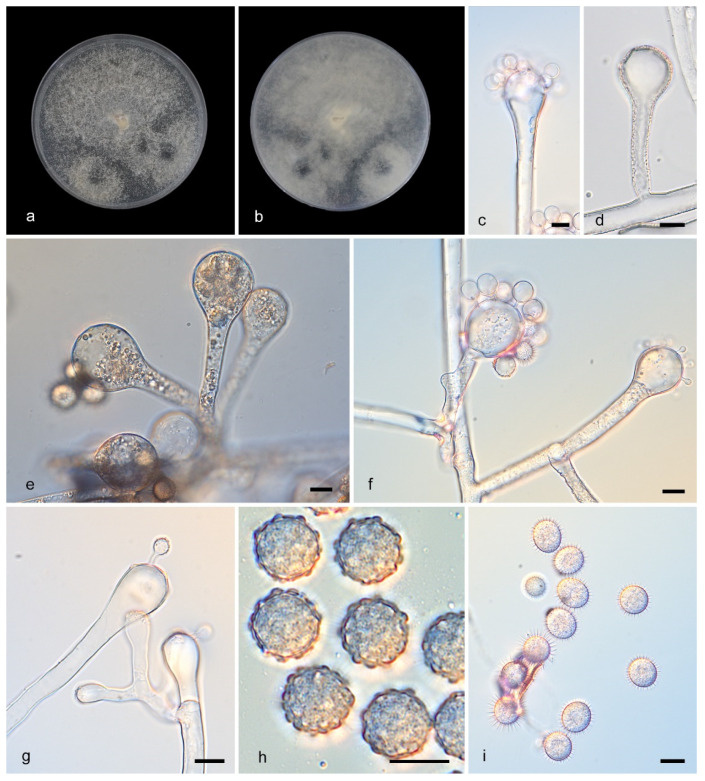
*Cunninghamella tuberculata* (CGMCC3.28889). (**a**,**b**): Colonies on PDA after 5 days of cultivation ((**a**) obverse, (**b**) reverse); (**c**–**g**): Sporangiophores showing branching patterns; (**h**,**i**): Sporangiola. Scale bars: 10 µm (**c**–**i**).

## 4. Discussion

The integrative approach, which combines molecular evidence with morphological, ecological, and other relevant data, forms the foundation for defining taxonomic novelties and reconstructing phylogenies for species delimitation [47,48,49,50,51,52]. Molecular sequences derived from type specimens provide indispensable data for constructing robust phylogenetic frameworks; these frameworks underpin evolutionarily meaningful classification systems [53]. Despite the primacy of molecular data, traditional morphological methods remain indispensable taxonomic tools [54]. Consequently, we describe novel species from China through integrated phylogenetic-morphological analyses [48,49,55].

This study resolves the evolutionary relationships among *Cunninghamella* species, including the eight new species reported herein via multilocus phylogenetics. We provide detailed morphological illustrations of key taxonomic features, including microscopic structures and colonial characteristics observed in pure cultures. We formally describe eight novel species: *C. crassior* sp. nov., *Cunninghamella diffundens* sp. nov., *C. fulvicolor* sp. nov., *C. fusca* sp. nov., *C. guttulata* sp. nov., *C. inaequalis* sp. nov., *C. monosporangiola* sp. nov., and *C. tuberculata* sp. nov. These species share the diagnostic morphological traits in *Cunninghamella*, echinulate sporangiola borne on verticillately branched vesicles. With these eight new species, this study expands the number of accepted species in *Cunninghamella* to 39.

## Figures and Tables

**Figure 1 microorganisms-13-02508-f001:**
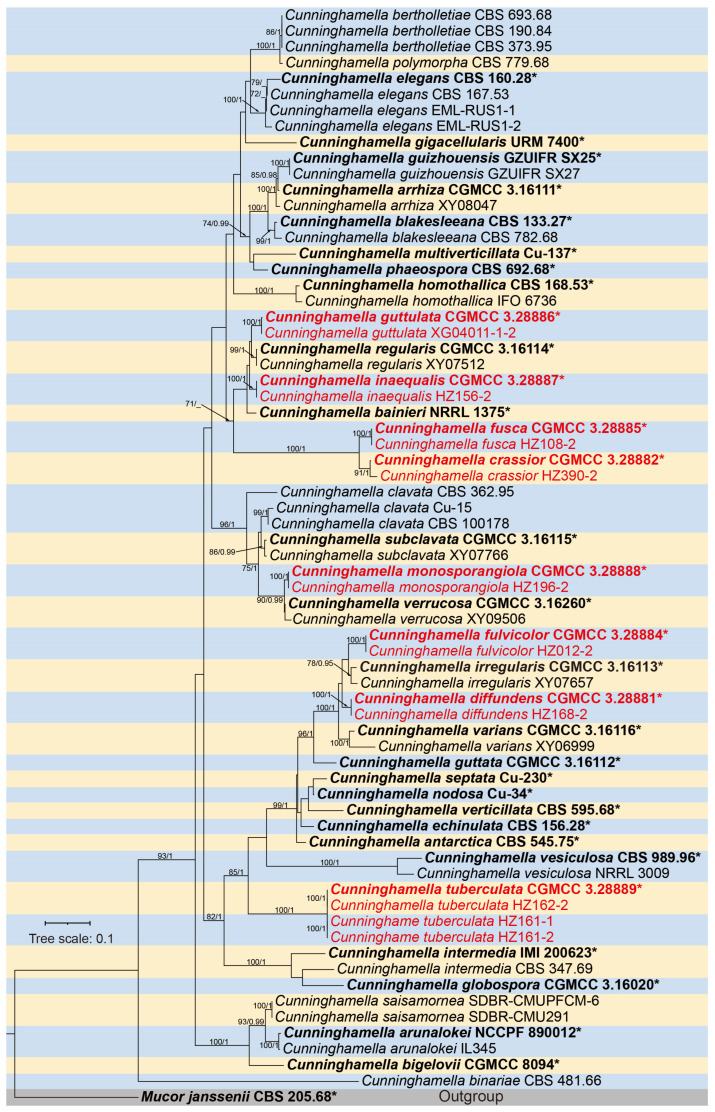
The Maximum Likelihood (ML) phylogenetic tree of the genus *Cunninghamella* based on ITS-LSU-*TEF1α* sequences. The ML Bootstrap Support (MLBS no less than 70%)/Bayesian Inference Posterior Probability (BIPP no less than 0.90) values are shown on branches. Strains newly isolated in this study are indicated in red. Ex-types are bold, plus an asterisk “*”. The scale bar above the tree roots represents 0.1 substitutions per site.

**Table 1 microorganisms-13-02508-t001:** PCR information used in this study.

Loci	PCR Primers	Primer Sequences (5′–3′)	PCR Cycles	References
ITS	ITS5	GGA AGT AAA AGT CGT AAC AAG G	94 °C 5 min; (94 °C 30 s, 55 °C 30 s, 72 °C 1 min) × 35 cycles; 72 °C 10 min	[40]
	ITS4	TCC TCC GCT TAT TGA TAT GC		
LSU	LR0R	GTA CCC GCT GAA CTT AAG C	94 °C 5 min; (94 °C 30 s, 51 °C 30 s, 72 °C 1 min) × 35 cycles; 72 °C 10 min	[41]
	LR7	TAC TAC CAC CAA GAT CT		
*TEF1α*	EF1-728F	ATG TGC AAG GCC GGT TTC GC	94 °C 5 min; (94 °C 30 s, 48 °C 30 s, 72 °C 1 min) × 35 cycles; 72 °C 10 min	[42,43]
	EF-2	GGA RGT ACC AGT SAT CAT GTT		

**Table 2 microorganisms-13-02508-t002:** GenBank accession numbers of *Cunninghamella* and *Mucor* sequences used in this study.

Species	Strains	GenBank Accession Numbers		
		ITS	LSU	*TEF1α*
** *C. antarctica* **	CBS 545.75 ^T^	JN205893	JN206597	KJ156492
*C. arrhiza*	CGMCC 3.16111 ^T^	OL678142	PQ399916	NA
	XY08047	OL678143	NA	NA
*C. arunalokei*	IL3459	MN431159	MN431158	NA
	NCCPF 890012 ^T^	NR_177485	NG153887	NA
*C. bainieri*	CBS 481.66	MH858865	MH870507	KJ156495
	CGMCC 8094 ^T^	KJ013403	KJ013405	KJ395944
	NRRL 1375 ^T^	AF254935	NA	NA
*C. bertholletiae*	CBS 190.84	JN205878	HM849701	NA
	CBS 373.95	JN205873	NA	KJ156497
	CBS 693.68	AF254931	MH870924	KJ156490
*C. bigelovii*	CGMCC 8094 ^T^	KJ013403	KJ013405	KJ395944
*C. binariae*	CBS 481.66	MH858865	MH870507	KJ156495
*C. blakesleeana*	CBS 133.27 ^T^	NR119974	MH866397	KJ156479
	CBS 782.68	JN205869	MH870950	KJ156478
*C. clavata*	CBS 100178	JN205890	JN206604	KJ156477
	Cu-15	AF254942	NA	NA
	CBS 362.95	JN205891	NA	NA
** *C. crassior* **	**CGMCC3.28882 ^T^**	**PV235924**	**PV239680**	**PV254891**
	**HZ390-2**	**PV235925**	**PV239681**	**PV254892**
** *C. diffundens* **	**CGMCC3.28881 ^T^**	**PV235920**	**PV239676**	**PV254889**
	**HZ168-2**	**PV235921**	**PV239677**	**PV254890**
*C. echinulata*	CBS 156.28 ^T^	JN205895	JN939199.1	KJ156500
*C. elegans*	CBS 160.28 ^T^	AF254928	NR_154747	KJ156470
	CBS 167.53	JN205882	HM849700	KJ156494
	EML-RUS1-1	MF806023	MF806027	NA
	EML-RUS1-2	MF806021	MF806028	NA
** *C. fulvicolor* **	**CGMCC3.28884 ^T^**	**PV235910**	**PV239666**	**PV254879**
	**HZ012-2**	**PV235911**	**PV239667**	**PV254880**
** *C. fusca* **	**CGMCC3.28885 ^T^**	**PV235908**	**PV239664**	**PV254877**
	**HZ108-2**	**PV235909**	**PV239665**	**PV254878**
*C. globospora*	CGMCC 3.16020 ^T^	MW264073	MW264132	NA
*C. gigacellularis*	URM 7400 ^T^	NR_168760	NG_068773	NA
*C. guizhouensis*	GZUIFR-SX25 ^T^	MN908596	MN908599	MN912633
	GZUIFR-SX27	MN908598	MN908601	MN912635
*C. guttata*	CGMCC 3.16112 ^T^	OL678144	PQ399917	NA
** *C. guttulata* **	**CGMCC3.28886 ^T^**	**PV235930**	**PV239686**	**PV254893**
	**XG04011-1-2**	**PV235931**	**PV239687**	**PV254894**
*C. homothallica*	CBS 168.53 ^T^	JN205863	JN206605	KJ156498
	IFO 6736	AF254941	NA	NA
** *C. inaequalis* **	**CGMCC3.28887 ^T^**	**PV235914**	**PV239670**	**PV254883**
	**HZ156-2**	**PV235915**	**PV239671**	**PV254884**
*C. intermedia*	CBS 347.69	JN205892	JN206606	NA
	IMI 200623 ^T^	AF254939	NA	NA
*C. irregularis*	CGMCC 3.16113 ^T^	OL678145	PQ399918	NA
	XY07657	OL678146	NA	NA
** *C. monosporangiola* **	**CGMCC3.28888 ^T^**	**PV235922**	**PV239678**	**PV785980**
	**HZ196-2**	**PV235923**	**PV239679**	**PV785981**
*C. multiverticillata*	CBS 989.96 ^T^	JN205897	HM849693	KJ156474
	Cu-137 ^T^	AF254933	NA	NA
	NRRL 3009	AF254943	NA	NA
*C. nodosa*	Cu-34 ^T^	AF346407	NA	NA
*C. phaeospora*	CBS 692.68 ^T^	JN205864	HM849697	NA
*C. polymorpha*	CBS 779.68	JN205874	JN206599	NA
*C. regularis*	CGMCC 3.16114 ^T^	OL678148	PQ399919	NA
	XY07512	OL678150	NA	NA
*C. saisamornae*	SDBR-CMUPFCM-6	MW709394	MW699571	MW715866
	SDBR-CMU291	MG571234	MW699591	MW715865
*C. septata*	Cu-230 ^T^	AF346408	NA	NA
*C. subclavata*	CGMCC 3.16115 ^T^	OL678152	NA	NA
	XY07766	OL678153	NA	NA
** *C. tuberculata* **	**CGMCC3.28889 ^T^**	**PV235918**	**PV239674**	**PV254887**
	**HZ162-2**	**PV235919**	**PV239675**	**PV254888**
*C. varians*	CGMCC 3.16116 ^T^	OL678154	PQ399920	NA
	XY06999	OL678155	NA	NA
*C. vesiculosa*	CBS 989.96 ^T^	JN205897	HM849693	KJ156474
	NRRL 3009	AF254943	NA	NA
*C. verrucosa*	CGMCC 3.16260	ON262555	ON261192	NA
	XY09506	ON262556	ON261193	NA
*C. verticillata*	CBS595.68 ^T^	AF254937	NA	NA
*M. janssenii*	CBS 205.68 ^T^	MH859119	MH870832	NA

**Notes**: New species discovered herein are shown in bold. The “T” indicates ex-type or ex-holotype strains. The “NA” stands for “not available”.

## Data Availability

No new data were created or analyzed in this study. Data sharing is not applicable to this article.

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
