# Peer review of "Unveiling Species Diversity Within Early-Diverging Fungi from China XI: Eight New Species of Cunninghamella (Mucoromycota)"

_microorganisms, 2025, doi:10.3390/microorganisms13112508_

Round 1
Reviewer 1 Report
Comments and Suggestions for Authors
The authors have done a good job reconstructing the phylogeny using several loci and two algorithms. Their results are generally well supported statistically, and these data are complemented by anatomical descriptions of microbiological cultures of these fungal species. Some formal aspects are detailed below in the line-by-line review. However, there are also some substantive issues that raise concerns regarding the overall quality of the work and the reliability of its results and conclusions.
First, it is necessary that the authors provide the Fungal Names before publication. It is essential that all new species be compared with other species of the genus, particularly with those that are morphologically most similar and phylogenetically closest. Many of the proposed new species are not compared with others in the genus, and when comparisons are made, they are rather superficial.
Finally, I do not agree that Cuninghamiella tuberculata and C. flocosibrunnea represent two different species. The four strains form a single monophyletic clade with no genetic distance between them, and the reported morphological characteristics are overlapping, since the colonies of C. tuberculata are floccose, grey-brown, with protrusions on the sporangiola, while C. flocosibrunnea is also floccose and grey-brown (line 181) and presents protrusions on its sporangiola (figure 4g–h), which the authors describe as spines.
For this reason, I am asking the authors to perform modifications specially in the Notes of species descriptions, adding some important information. And reduce the number of new species from 9 to 8.
L 15-16 The sequencies/analysis were done in the current work? If yes avoid to use the word studies, since it seems like if the species were discovered in this and other works. You can change the word studies by analysis.
L 14-28 The species names can be abbreviated since all are from the same genus maintaining only the initial from the 2nd to the 9th species. In addition should be added the taxonomical authority of the species in the first time that the species are writed in within the text.
L 30 The Introduction is too bried, failing to build a conceptual framework that helps the reader to understand the state of the art, and why the authors performthis work.
L 31 Add taxonomic authority à Cunninghamella, Matruchot.
L 36 I Suggest to change the name of the country by it´s more formal one Czech Republic.
L 42-46 These lines are grammatically unconexed
L 47-53 This paragraph didn´t explained why the authors studied Cunninghamella, why in this territory, which was the objective… The new species were part of a species complex? Did you had some indication or evidence that suggest that hidden diversity within this genus was present in the territory? As this paragraph is written looks like conclusions instead of Introduction.
L 89 sequences were obtained from nucleotide Data base from NCBI, BLAST is the tool that checks from the database the most similar sequences. You can directly download all the sequences from the same genus from the database without Blasting it before, however you can not obtain the sequences if you don’t download it from nucleotide. Modify the sentence in accordance.
Blast is not offering you the information about Ex-types and you selected it. Write how you selected in fact the species.
L 109-110 lower left corner?
L 112 Why you used Mucor janssenii as outgroup, can you justify how you selected this taxon instead others? And even mor why only one species and not two or three different taxa (it improves the quality)? I think that it should be specified in MM.
L 113-114 Why you don´t refer as base pairs instead of characters or positions? Is because you are considering the gaps and not all the positions/characters corresponds to a base? I consider that the most correct is to refer to base pairs.
L 116 It is not explained how you choose the substitution models? You calculated it? Which software and specifications did you used? Or you used GTR+G and GTR+G+I based in some reference? You should specify it. The models employed are duplicated, to my opinion if you inferred it, how you inferred should be in MM and the chosen models in results.
L 117 Part of the results, number of generations are more feasible to be in MM section.
L 121 I suggest to rephrase “Nine monographic clades were formed” which suggests intentional construction rather than objective inference. Phylogenetic analyses reveal relationships, they do not create them. A more accurate formulation would be “The phylogenetic analysis showed nine monographic clades.”
To propose the new species, the species of the genus should be revised and morphology should be compared. Did you check against which species or cultures?
L 127 This species is not compared with the related species of the genus, you should do it (also for the rest of the species) compare morphology attending to special characters like septum or sporangia with similar species and how to differentiate it from closest species (phylogenetic) or most similar (morphologically).
L 128 Should be applied before acceptation (For this and the rest of the species).
L 163-168 The morphological comparison should be quite more extensive, not only one character and not only one species.
L 177 brunnea means brown color in latin, for the yellow color the commonly used is flava/flavo. I suggest to revise it and modify the word/color or if it is a yellow tone close to brown refer to the specific color tone. I suggest rename as floccosiflava or describe as Brown color.
L 189-192 The wide of these lines is different, which is the color of the colonies? In the previous lines you said brown, now you said yellow. And you should compare the new species with any other relevant species at least those similar morphologically and the closest relatves phylogenetically.
L 349 Molecular evidence is not the only evidence and should be complemented with other lines of evidence in an integrative taxonomy approach.
L 350 Your phylogenetic tree is not a species tree, correct since yours is not a species phylogeny it is the phylogeny of the genus. The periphrasis “reconstructing species phylogenies” seems incorrect, change it to reconstructing phylogenies, or reconstructing phylogenies for species delimitation.
L 355 this approach is called Integrative taxonomy, you should add a reference here, for example Alors et al. 2016 PLoS ONE 11(2), or other reference in a more related taxa.
Author Response
First, it is necessary that the authors provide the Fungal Names before publication. It is essential that all new species be compared with other species of the genus, particularly with those that are morphologically most similar and phylogenetically closest. Many of the proposed new species are not compared with others in the genus, and when comparisons are made, they are rather superficial.
Already applied for Fungal Names and made revisions in the manuscript.
Finally, I do not agree that Cuninghamiella tuberculata and C. flocosibrunnea represent two different species. The four strains form a single monophyletic clade with no genetic distance between them, and the reported morphological characteristics are overlapping, since the colonies of C. tuberculata are floccose, grey-brown, with protrusions on the sporangiola, while C. flocosibrunnea is also floccose and grey-brown (line 181) and presents protrusions on its sporangiola (figure 4g–h), which the authors describe as spines.
The reviewers' comments have been accepted, and Cunninghamella floccosibrunnea is recognized as the same species as C. tuberculata.
L 15-16 The sequencies/analysis were done in the current work? If yes avoid to use the word studies, since it seems like if the species were discovered in this and other works. You can change the word studies by analysis.
Make revisions in the manuscript
L 14-28 The species names can be abbreviated since all are from the same genus maintaining only the initial from the 2nd to the 9th species. In addition should be added the taxonomical authority of the species in the first time that the species are writed in within the text.
Make revisions in the manuscript
L 30 The Introduction is too bried, failing to build a conceptual framework that helps the reader to understand the state of the art, and why the authors performthis work.
Make revisions in the manuscript
L 31 Add taxonomic authority à Cunninghamella, Matruchot.
Make revisions in the manuscript
L 36 I Suggest to change the name of the country by it´s more formal one Czech Republic.
Make revisions in the manuscript
L 42-46 These lines are grammatically unconexed
Make revisions in the manuscript
L 47-53 This paragraph didn´t explained why the authors studied Cunninghamella, why in this territory, which was the objective… The new species were part of a species complex? Did you had some indication or evidence that suggest that hidden diversity within this genus was present in the territory? As this paragraph is written looks like conclusions instead of Introduction.
Make revisions in the manuscript
L 89 sequences were obtained from nucleotide Data base from NCBI, BLAST is the tool that checks from the database the most similar sequences. You can directly download all the sequences from the same genus from the database without Blasting it before, however you can not obtain the sequences if you don’t download it from nucleotide. Modify the sentence in accordance.
Make revisions in the manuscript
Blast is not offering you the information about Ex-types and you selected it. Write how you selected in fact the species.
Based on previous research results, the GeneBank tables were organized in the published literature. eg,Ding(https://doi.org/10.3390/jof11060417)and Zhao(https://doi.org/10.1080/12298093.2021.1904555)
L 109-110 lower left corner?
Make revisions in the manuscript
L 112 Why you used Mucor janssenii as outgroup, can you justify how you selected this taxon instead others? And even mor why only one species and not two or three different taxa (it improves the quality)? I think that it should be specified in MM.
- Based on previous research results,eg,Ding(https://doi.org/10.3390/jof11060417)and Zhao(https://doi.org/10.1080/12298093.2021.1904555)
- Continuing the previous approach, in order to facilitate direct comparison with a large number of prior studies
- Already supplemented
L 113-114 Why you don´t refer as base pairs instead of characters or positions? Is because you are considering the gaps and not all the positions/characters corresponds to a base? I consider that the most correct is to refer to base pairs.
The comparison process has gaps
L 116 It is not explained how you choose the substitution models? You calculated it? Which software and specifications did you used? Or you used GTR+G and GTR+G+I based in some reference? You should specify it. The models employed are duplicated, to my opinion if you inferred it, how you inferred should be in MM and the chosen models in results.
We used PAPU software to select the optimal surrogate model for each gene partition sequence, and the software parameters have been added to the manuscript
L 117 Part of the results, number of generations are more feasible to be in MM section.
Adjustments have been made in the manuscript
L 121 I suggest to rephrase “Nine monographic clades were formed” which suggests intentional construction rather than objective inference. Phylogenetic analyses reveal relationships, they do not create them. A more accurate formulation would be “The phylogenetic analysis showed nine monographic clades.”
Make revisions in the manuscript
To propose the new species, the species of the genus should be revised and morphology should be compared. Did you check against which species or cultures?
- Cunninghamella globospora (https://doi.org/10.1080/12298093.2021.1904555),
- Cunninghamella arrhiza, Cunninghamella guttata, Cunninghamella irregularis, Cunninghamella nodosa, Cunninghamella regularis, Cunninghamella subclavata, Cunninghamella subclavata (https://doi.org/10.1007/s13225-023-00525-4)
L 127 This species is not compared with the related species of the genus, you should do it (also for the rest of the species) compare morphology attending to special characters like septum or sporangia with similar species and how to differentiate it from closest species (phylogenetic) or most similar (morphologically).
Compared with similar species, C. crassior has clustered on a separate branch of the evolutionary tree and can be identified as a new species
L 128 Should be applied before acceptation (For this and the rest of the species).
Already applied, in the process of revising the manuscript
L 163-168 The morphological comparison should be quite more extensive, not only one character and not only one species.
Make revisions in the manuscript
L 177 brunnea means brown color in latin, for the yellow color the commonly used is flava/flavo. I suggest to revise it and modify the word/color or if it is a yellow tone close to brown refer to the specific color tone. I suggest rename as floccosiflava or describe as Brown color.
Accept the feedback and confirm that Cunninghamella floccosibrunnea and Cunninghamella tuberculata are the same species. Revise the manuscript accordingly
L 189-192 The wide of these lines is different, which is the color of the colonies? In the previous lines you said brown, now you said yellow. And you should compare the new species with any other relevant species at least those similar morphologically and the closest relatves phylogenetically.
Accept the feedback and confirm that Cunninghamella floccosibrunnea and Cunninghamella tuberculata are the same species. Revise the manuscript accordingly
L 349 Molecular evidence is not the only evidence and should be complemented with other lines of evidence in an integrative taxonomy approach.
Make revisions in the manuscript
L 350 Your phylogenetic tree is not a species tree, correct since yours is not a species phylogeny it is the phylogeny of the genus. The periphrasis “reconstructing species phylogenies” seems incorrect, change it to reconstructing phylogenies, or reconstructing phylogenies for species delimitation.
Make revisions in the manuscript
L 355 this approach is called Integrative taxonomy, you should add a reference here, for example Alors et al. 2016 PLoS ONE 11(2), or other reference in a more related taxa.
Relevant literature has been added to the manuscript
Reviewer 2 Report
Comments and Suggestions for Authors
This is a valuable manuscript. A morphological and phylogenetic description of nine new species of the genus Cunninghamella is presented. Generally, these species exhibit characteristics that significantly distinguish them from each other. Extensive microscopic photographic documentation is provided. In addition to typographical errors, the current version also contains a large number of errors in the descriptions of new species. Examples are provided in the Comments. Taxonomic work requires precise data, there can be no contradictions and ambiguities. The current version requires a major revision.
Comments
Line 30 Introduction should be supplemented with a paragraph devoted to the characteristic features of the genus Cunninghamella. Various features are given in various places in the manuscript, which are supposedly typical of this genus; this needs to be systematized.
Line 34: It should be carriers [1].
Line 38: Matruchot in 1903 [15] – this text requires correction. In the References, this publication is numbered [18].
Line 48: Liu and Zhao [14, 27] – there is an error here; it does not match the data in the References.
Line 50: Cunninghamella – it should be italicized.
Chapter 3.2.2 to 3.2.9: Fungal names should be italicized.
Line 106: The names of some fungi are misspelled in Figure 1 – this requires correction!
Lines 143–144, this text repeated at the end of every chapter in Taxonomy is redundant and should be removed everywhere.
Line 177 brunnea = yellow? This requires correction or clarification.
Line 177 ‘The epithet “floccosibrunnea” describes floccose yellow colonies.’ However, in Line 181 it says ‘floccose, initially white, soon turning gray, and finally becoming gray-brown, in reverse gray.’ This is an error that needs to be corrected. This indicates that the epithet “floccosibrunnea” is unjustified.
Line 188 ‘Chlamydospores observed.’ – their characteristics should be provided.
Line 190 ‘These two strains produce yellow floccose colonies, a distinctive feature in Cunninghamella.’ – this is inconsistent with the description given for this species.
Line 197 sp.nov. Fig. 5 – This requires correction.
Line 199 refers to the yellowish-brown pigmentation of colonies. Why is this characteristic not mentioned at all in the colony description below? Why is this pigmentation not visible in the colony photos?
Line 216 ‘These two strains form yellowish-brown pigmented colonies’ – This characteristic is not mentioned in the colony characteristics. More details are needed. Why is this pigment not visible in the photo?
Line 224 sp. nov. Fig. 6 – This requires minor correction.
Line 240 ‘…. produce brown sporangiola, while C. crassior produces brownish ones.” Unfortunately, this feature was not given in the description of Cunninghamella crassior (chapter 3.2.1.).
Line 247 it should be Fig. 7
Line 249 teardrop-shaped sporangiola. This feature is not given in the species description. A different shape of sporangiola is given in the description
Line 263 drop-shaped spines Chlamydospores – this text requires correction
Line 267 ‘produce teardrop-shaped sporangiola, a distinctive feature in the genus Cuninghamella’ – This text is, in my opinion, incorrect and requires correction. Please read your own descriptions of sporangiola for individual species
Line 273 h. sporangiophores – this is an error; this photo does not show sporangiophores
Line 275 it should be Fig. 8 (this should be corrected throughout the manuscript)
Line 301 single sporangiola – this text requires clarification. This feature should be added to the species description.
Line 321 I, j – this requires correction.
Author Response
Line 30 ntroduction should be supplemented with a paragraph devoted to the characteristic features of the genus Cunninghamella. Various features are given in various places in the manuscript, which are supposedly typical of this genus; this needs to be systematized.
Add to the introduction section of the manuscript
Line 34: It should be carriers [1].
Modifications have been made
Line 38: Matruchot in 1903 [15] – this text requires correction. In the References, this publication is numbered [18].
Modifications have been made
Line 48: Liu and Zhao [14, 27] – there is an error here; it does not match the data in the References.
Modifications have been made
Line 50: Cunninghamella – it should be italicized.
Modifications have been made
Chapter 3.2.2 to 3.2.9: Fungal names should be italicized.
Modifications have been made
Line 106: The names of some fungi are misspelled in Figure 1 – this requires correction!
Modifications have been made
Lines 143–144, this text repeated at the end of every chapter in Taxonomy is redundant and should be removed everywhere.
Modifications have been made
Line 177 brunnea = yellow? This requires correction or clarification.
The opinions of other reviewers have already been accepted, recognizing Cunninghamella floccosibrunnea and C. tuberculata as the same species.
Line 177 ‘The epithet “floccosibrunnea” describes floccose yellow colonies.’ However, in Line 181 it says ‘floccose, initially white, soon turning gray, and finally becoming gray-brown, in reverse gray.’ This is an error that needs to be corrected. This indicates that the epithet “floccosibrunnea” is unjustified.
The opinions of other reviewers have already been accepted, recognizing Cunninghamella floccosibrunnea and C. tuberculata as the same species.
Line 188 ‘Chlamydospores observed.’ – their characteristics should be provided.
Modifications have been made
Line 190 ‘These two strains produce yellow floccose colonies, a distinctive feature in Cunninghamella.’ – this is inconsistent with the description given for this species.
The opinions of other reviewers have already been accepted, recognizing Cunninghamella floccosibrunnea and C. tuberculata as the same species.
Line 197 sp.nov. Fig. 5 – This requires correction.
Modifications have been made
Line 199 refers to the yellowish-brown pigmentation of colonies. Why is this characteristic not mentioned at all in the colony description below? Why is this pigmentation not visible in the colony photos?
Modifications have been made,yellow precipitate can be observed at the center of the colony in the image
Line 216 ‘These two strains form yellowish-brown pigmented colonies’ – This characteristic is not mentioned in the colony characteristics. More details are needed. Why is this pigment not visible in the photo?
Modifications have been made
Line 224 sp. nov. Fig. 6 – This requires minor correction.
Modifications have been made
Line 240 ‘…. produce brown sporangiola, while C. crassior produces brownish ones.” Unfortunately, this feature was not given in the description of Cunninghamella crassior (chapter 3.2.1.).
Modifications have been made
Line 247 it should be Fig. 7
Modifications have been made
Line 249 teardrop-shaped sporangiola. This feature is not given in the species description. A different shape of sporangiola is given in the description
Modifications have been made
Line 263 drop-shaped spines Chlamydospores – this text requires correction
Modifications have been made
Line 267 ‘produce teardrop-shaped sporangiola, a distinctive feature in the genus Cuninghamella’ – This text is, in my opinion, incorrect and requires correction. Please read your own descriptions of sporangiola for individual species
Modifications have been made
Line 273 h. sporangiophores – this is an error; this photo does not show sporangiophores
Modifications have been made
Line 275 it should be Fig. 8 (this should be corrected throughout the manuscript)
Modifications have been made
Line 301 single sporangiola – this text requires clarification. This feature should be added to the species description.
Modifications have been made
Line 321 I, j – this requires correction
Modifications have been made
Round 2
Reviewer 2 Report
Comments and Suggestions for Authors
Unfortunately, proper corrections were not made in all the indicated places in review 1. It is particularly important that there are errors in the names of fungi (I have pointed them out below in the Comments). The entire text requires careful reading, as in places where changes were made, the new text is incorrect. The manuscript requires minor revision.
Comments
Lines 21-22 text urgently requires correction.
Line 43-44 text urgently requires correction.
Line 111 it should be Lendn.
Line 113 the text requires a small correction.
Liner 157 For each species, a 'Fungal Name: e.g., FN573054' is given. What does this mean? This is a fungal number, not a fungal name.
Figure 1 "Cunninghame flocosibrunnea". This name is written twice, which is an error. This species has been removed
Figure 1 here is an incorrect name for the fungus: it should be 'Cunninghamella monosporangiola' instead of 'Cunninghamella monosporella'.
Line 206 text urgently requires correction.
Line 200-207 the entire paragraph requires correction. It's unclear why the name Cunninghamella bispora, a fungus that hasn't been studied at all, is listed here (this is probably an error; it should be Cunninghamella diffundens – twice in this paragraph).
Line 210 – 213: The text needs to be corrected.
Line 266: Produces irregular spores – provide the exact name of the spore type.
Line 275: There's an error in the name of the fungus; it should be 'C. irregularis' instead of 'C. issegularis'.
Line 338: Should be 'sporangiola' – this is the case elsewhere.
Author Response
Dear reviewer:
Thank you for the reviewer's comments on the article. The revisions have been made and marked in the manuscript. Upload the manuscript in the attachment.
Lines 21-22 text urgently requires correction.
Modifications have been made in the manuscript
Line 43-44 text urgently requires correction.
Modifications have been made in the manuscript
Line 111 it should be Lendn.
Modifications have been made in the manuscript
Line 113 the text requires a small correction.
Modifications have been made in the manuscript
Liner 157 For each species, a 'Fungal Name: e.g., FN573054' is given. What does this mean? This is a fungal number, not a fungal name.
This is an application for a fungal number on the Fungal Names website: https://nmdc.cn/fungalnames/
Figure 1 "Cunninghame flocosibrunnea". This name is written twice, which is an error. This species has been removed
Modified in Figure 1
Figure 1 here is an incorrect name for the fungus: it should be 'Cunninghamella monosporangiola' instead of 'Cunninghamella monosporella'.
Modified in Figure 1
Line 206 text urgently requires correction.
Modifications have been made in the manuscript
Line 200-207 the entire paragraph requires correction. It's unclear why the name Cunninghamella bispora, a fungus that hasn't been studied at all, is listed here (this is probably an error; it should be Cunninghamella diffundens – twice in this paragraph).
Modifications have been made in the manuscript
Line 210 – 213: The text needs to be corrected.
Modifications have been made in the manuscript
Line 266: Produces irregular spores – provide the exact name of the spore type.
Modifications have been made in the manuscript
Line 275: There's an error in the name of the fungus; it should be 'C. irregularis' instead of 'C. issegularis'.
Modifications have been made in the manuscript
Line 338: Should be 'sporangiola' – this is the case elsewhere.
Modifications have been made in the manuscript